# CONTRASTIVE ESTIMATION REVEALS TOPIC POSTERIOR INFORMATION TO LINEAR MODELS

## ABSTRACT

Contrastive learning is an approach to representation learning that utilizes naturally occurring similar and dissimilar pairs of data points to find useful embeddings of data. In the context of document classification under topic modeling assumptions, we prove that contrastive learning is capable of recovering a representation of documents that reveals their underlying topic posterior information to linear models. We apply this procedure in a semi-supervised setup and demonstrate empirically that linear classifiers with these representations perform well in document classification tasks with very few training examples.

## 1 INTRODUCTION

Using unlabeled data to find useful embeddings is a central challenge in representation learning. Classical approaches to this task often start by fitting some type of structure to the unlabeled data, such as a generative model or a dictionary, and then embed future data via inference with the fitted structure (Blei et al., 2003; Raina et al., 2007). While principled, this approach is not without its drawbacks. One issue is that learning structures and performing inference is often hard in general (Sontag & Roy, 2011; Arora et al., 2012). Another issue is that we must a priori choose a structure and method for fitting the unlabeled data, and unsupervised methods for learning these structures can be sensitive to model misspecification (Kulesza et al., 2014).

Contrastive learning (also called noise contrastive estimation, or NCE) is an alternative representation learning approach that tries to capture the latent structure in unlabeled data implicitly. At a high level, these methods formulate a classification problem in which the goal is to distinguish examples that naturally occur in pairs, called positive samples, from randomly paired examples, called negative samples. The particular choice of positive samples depends on the setting. In image representation problems, for example, patches from the same image or neighboring frames from videos may serve as positive examples (Wang & Gupta, 2015; Hjelm et al., 2018). In text modeling, the positive samples may be neighboring sentences (Logeswaran & Lee, 2018; Devlin et al., 2018). The idea is that in the course of learning to distinguish between semantically similar positive examples and randomly chosen negative examples, we will capture some of the latent semantic information.

In this work, we look "under the hood" of contrastive learning and consider its application to document modeling, where the goal is to construct useful vector representations of text documents in a corpus. In this setting, there is a natural source of positive and negative examples: a positive example is simply a document from the corpus, and a negative example is one formed by pasting together the first half of one document and the second half of another (independently chosen) document. We prove that when the corpus is generated by a topic model, learning to distinguish between these two types of documents yields representations that are closely related to their underlying latent variables.

One potential application of contrastive learning is in a semi-supervised setting, where there is a small amount of labeled data as well as a much larger collection of unlabeled data. In these situations, purely supervised methods that fit complicated models may have poor performance due to the limited amount of labeled data. On the other hand, when the labels are well-approximated by some function of the latent structure, our results show that an effective strategy is to fit linear functions, which may be learned with relatively little labeled data, on top of contrastive representations. In our experiments, we verify empirically that this approach produces reasonable results.

**Contributions.** The primary goal of this work is to shed light on what contrastive learning techniques uncover in the presence of latent structure. To this end, we focus on the setting of document modeling where latent structure is induced by a topic model. Here, our contrastive learning objective is to distinguish true documents from 'fake' documents that are composed by randomly pasting together two document halves from the corpus. We consider two types of architectures or functional forms of solutions for this problem, both trained with logistic loss.

The first architecture, on which our theoretical analysis will focus, consists of general functions of the form $f(\cdot, \cdot)$. Here, we have trained $f$ so that $f(x, x')$ indicates the confidence of the model that $x$ and $x'$ are two halves of the same document. To embed a new document $x$ using $f$, we propose a *landmark embedding* procedure: fix documents $l_1, \ldots, l_M$ (our so-called landmarks) and create the embedding $\phi(x)$ using a function of the predictions $f(x, l_1), \ldots, f(x, l_M)$. In Section 4, we show that the embedding $\phi(x)$ is a linear transformation of the underlying topic posterior moments of $x$. Moreover, under certain conditions this linear relationship is invertible, so that linear functions of $\phi(x)$ correspond to polynomial functions of the topic posterior of document $x$. In Section 5, we show that errors in $f$ on the contrastive learning objective transfer smoothly to errors in $\phi(x)$ as a linear transformation of the topic posterior of $x$. Thus, as the quality of $f$ improves, linear functions of $\phi(x)$ more closely approximate polynomial functions of the topic posterior of document $x$.

Unfortunately, the landmark embedding can require quite a few landmarks before our theoretical results kick in. Moreover, embedding a document requires $M$ evaluations of $f$, which can be expensive. To circumvent this, in Section 7 we introduce a *direct embedding* procedure that more closely matches what is done in practice. We use an architecture of the form $f_1(x)^\top f_2(x')$ where $f_1, f_2$ are functions with $d$-dimensional outputs, and we train this architecture on the same contrastive learning task as before. To embed a document $x$, we simply use the evaluation $f_1(x)$. In Section 7, we evaluate this embedding on a semi-supervised learning task, and we show that it has reasonable performance. Indeed, the direct embedding method outperforms the landmark embedding method, which raises the question of whether or not anything can be theoretically proven about the direct embedding method. We leave this question to future work.

**Related work.** Reducing an unsupervised problem to a synthetically-generated supervised problem is a well-studied technique. In dynamical systems modeling, Langford et al. (2009) showed that the solutions to a few forward prediction problems can be used to track the underlying state of a non-linear dynamical system. For linear dynamics, the idea is also seen in autoregressive models (Yule, 1927). In anomaly/outlier detection, a useful technique is to learn a classifier that distinguishes between true samples from a distribution and fake samples from some synthetic distribution (Steinwart et al., 2005; Abe et al., 2006). Similarly, estimating the parameters of a probabilistic model can be reduced to learning to classify between true data and randomly generated noise (Gutmann & Hyvärinen, 2010).

In the context of natural language processing, methods such as skip-gram and continuous bag-of-words turn the problem of finding word embeddings into a prediction problem (Mikolov et al., 2013a;b). Modern language representation training algorithms such as BERT and QT also use naturally occurring classification tasks such as predicting randomly masked elements of a sentence or discriminating whether or not two sentences are adjacent (Devlin et al., 2018; Logeswaran & Lee, 2018). Training these models often employs a technique called negative sampling, in which softmax prediction probabilities are estimated by randomly sampling examples; this bears close resemblance to the way that negative examples are produced in contrastive learning.

Most relevant to the current paper, Arora et al. (2019) gave a theoretical analysis of contrastive learning. They considered the specific setting of trying to minimize the contrastive loss

$$L(f) = \mathbb{E}_{x, x_+, x_-} [\ell \left( f(x)^\top (f(x_+) - f(x_-)) \right)]$$

where $(x, x_+)$ is a positive pair and $(x, x_-)$ is a negative pair. They showed that if there is an underlying collection of latent classes and positive examples are generated by draws from the same class, then minimizing the contrastive loss over embedding functions $f$ yields good representations for the classification task of distinguishing latent classes.

The main difference between our work and that of Arora et al. (2019) is that we adopt a generative modeling perspective and induce the contrastive distribution naturally, while they do not make generative assumptions but assume the contrastive distribution is directly induced by the downstream

---

**Algorithm 1** Contrastive Estimation with Documents

---

**Input:** Corpus $\mathcal{U}$ of unlabeled documents. **Initialize:** $S = \emptyset$.
**for** $i = 1, \ldots, n$ **do**

    Sample $x$ and $\tilde{x}$ independently from unif($\mathcal{U}$); $S \leftarrow S \cup \begin{cases} \{(x^{(1)}, x^{(2)}, 1)\} \text{ w.p. } 1/2 \\ \{(x^{(1)}, \tilde{x}^{(2)}, 0)\} \text{ w.p. } 1/2 \end{cases}$

**end for**
Solve the optimization problem

$$\hat{f} = \underset{f}{\text{minimize}} \sum_{(x^{(1)}, x^{(2)}, y) \in S} y \log\left(1 + e^{-f(x^{(1)}, x^{(2)})}\right) + (1 - y) \log\left(1 + e^{f(x^{(1)}, x^{(2)})}\right)$$

Select landmark documents $l_1, \ldots, l_M$ and embed $\hat{\phi}(x) = \left(\exp\left(\hat{f}(x, l_i)\right) : i \in [M]\right)$.

---

classification task. In particular, our contrastive distribution and supervised learning problem are only *indirectly* related through the latent variables in the generative model, while Arora et al. assume an explicit connection. The focus of our work is therefore complementary to theirs: we study the types of functions that can be succinctly expressed with the contrastive representation in our generative modeling setup. In addition, our results apply to semi-supervised regression, but it is unclear how to define their contrastive distribution in this setting; this makes it difficult to apply their results here. Finally, Arora et al. point out the method they study has limitations that arise when the number of latent classes is small and the probability of negative samples having the same class is high. In our setting, class collisions turn out not to be a problem since our embeddings explicitly utilize conditional probability information from the solution to our contrastive learning objective.

## 2 SETUP

Let $\mathcal{V}$ denote a finite vocabulary. A *topic* is a distribution over $\mathcal{V}$. We will assume that we have $K$ such topics, and denote the corresponding distributions as $O(\cdot \mid k)$ for $k = 1, \ldots, K$. To generate a length $m$ document $x$, one first draws a vector $w$ from $\Delta^K$, the $K$-dimensional probability simplex, and then samples each of the $m$ words $x_1, \ldots, x_m$ by first sampling the latent variable $z_i \sim w$ and drawing $x_i \sim O(\cdot \mid z_i)$. We note that documents are allowed to take different lengths.

We will also be interested in the case where each document has an associated label $\ell \in \mathbb{R}$. One natural restriction to make on a label is that it is conditionally independent of the document given the topic distribution of the document. Thus, we will assume that there is a joint distribution $\mathcal{D}$ of triples $(x, w, \ell)$, where $(x, w)$ are generated according to the topic model described above, and then $\ell$ is drawn from some distribution conditioned on $w$. One of the goals of this paper is to characterize the functional forms of this conditional distribution that are most suited to contrastive learning.

In the representation learning approach to the semi-supervised setting, we are given a large collection $\mathcal{U}$ of documents with no labels, and a small collection $\mathcal{L}$ of labeled documents. Using $\mathcal{U}$, we learn a feature map $\hat{\phi}$ that will form the basis of our predictions. Then, using $\mathcal{L}$, we learn a simple predictor based on $\hat{\phi}$, such as a linear function, to predict the label $\ell$ given $\hat{\phi}(x)$.

## 3 CONTRASTIVE LEARNING ALGORITHM

In contrastive learning, examples come in the form of similar and dissimilar pairs of points, where the exact definition of similar/dissimilar depends on the task at hand. Our construction of similar pairs will take the form of randomly splitting a document into two documents, and our dissimilar pairs will consist of subsampled documents from two randomly chosen documents. In the generative modeling setup, since the words are i.i.d. conditional on the topic distribution, a natural way to split a document $x$ into two is to call the first half of the words $x^{(1)}$ and the second half $x^{(2)}$. In our experiments, we split the documents by applying a random permutation to the word tokens and partitioning in half, thus effectively ignoring the word ordering (as is common in topic models).

The contrastive representation learning procedure is displayed in Algorithm 1. It uses a finite-sample approximation to the contrastive distribution $\mathcal{D}_{\text{contrast}}$ described as follows: (a) sample a document $x$ and partition it into $(x^{(1)}, x^{(2)})$, (b) with probability $1/2$ output $(x^{(1)}, x^{(2)}, 1)$, (c) with probability $1/2$, sample a second document $(\tilde{x}^{(1)}, \tilde{x}^{(2)})$ and output $(x^{(1)}, \tilde{x}^{(2)}, 0)$. For $(x, x', y) \sim \mathcal{D}_{\text{contrast}}$, the parts $x$ and $x'$ are the two halves of a (possibly synthetic) document, and $y$ is the binary label. Our contrastive learning objective is to minimize the binary cross-entropy loss of discriminating between positive and negative examples:

$$L_{\text{contrast}}(f) := \mathbb{E}_{(x,x',y)\sim\mathcal{D}_{\text{contrast}}}\left[ y \log\left(1 + e^{-f(x,x')}\right) + (1-y)\log\left(1 + e^{f(x,x')}\right)\right]. \quad (1)$$

In our algorithm, we approximate this expectation via sampling and optimize the empirical objective, which yields an approximate minimizer $\hat{f}$ (chosen from some function class $\mathcal{F}$).

To see why optimizing this contrastive learning objective is so useful, let $f^\star$ be the global minimizer of Eq. (1). By Bayes' theorem we have that $g^\star := \exp(f^\star)$ satisfies the following:

$$g^\star(x, x') := \exp(f^\star(x, x')) = \frac{\mathbb{P}(y = 1 \mid x, x')}{\mathbb{P}(y = 0 \mid x, x')} = \frac{\mathbb{P}(x^{(1)} = x, x^{(2)} = x')}{\mathbb{P}(x^{(1)} = x)\mathbb{P}(x^{(2)} = x')}.$$

Thus, $g^\star(x, x')$ captures the ratio of the probability of $x$ and $x'$ co-occurring as the first and second halves of the same document and the product of their marginal probabilities.

In Eq. (1), we have not imposed any constraints on the functions over which we are optimizing. Thus, we seek to extract a useful embedding from $g^\star$ using only *black box access* to $g^\star$. Our approach to this problem is to select some set of fixed documents, which we call *landmarks*, and to embed by utilizing the predictions of $g^\star$ on these landmarks.

Formally, we select documents $l_1, \ldots, l_M$ and represent document $x$ as[1]

$$\phi^\star(x) := (g^\star(x, l_1), \ldots, g^\star(x, l_M)). \quad (2)$$

This yields the final document-level representation, which can be used for downstream tasks. As we shall see in Section 4, when the documents have an underlying topic structure, $\phi^\star(x)$ is related to the posterior information of the topics by a linear transformation and this linear transformation is invertible whenever the landmarks $l_1, \ldots, l_M$ are sufficiently diverse.

In practice, we only have access to an approximate minimizer $\hat{f}$ of Eq. (1). Thus, our embedding in practice will be given by

$$\hat{\phi}(x) = \left(\exp\left(\hat{f}(x, l_1)\right), \ldots, \exp\left(\hat{f}(x, l_M)\right)\right).$$

In Section 5 we will see that, under some mild assumptions, our claims about $\phi^\star$ also hold true for $\hat{\phi}$ up to some small errors.

Finally, we point out that there is nothing special about the binary cross-entropy loss. We may replace this loss in Eq. (1) with any proper scoring rule (Shuford et al., 1966; Buja et al., 2005), so long as the appropriate non-linear transformation is applied to the resulting predictions.

## 4    RECOVERING TOPIC STRUCTURE

In this section, we focus on expressivity of the contrastive representation, showing that polynomial functions of the topic posterior can be represented as *linear* functions of the representation. To do so, we ignore statistical issues and assume that we have access to the oracle representations $g^\star(x, \cdot)$. In Section 5, we address statistical issues.

Recall the generative topic model process for a document $x$. We first draw a topic vector $w \in \Delta^K$. Then for each word $i = 1, \ldots, \text{length}(x)$, we draw $z_i \sim \text{Categorical}(w)$ and $x_i \sim O(\cdot \mid z_i)$. We will show that when documents are generated according to the above model, the embedding of a document $x$ in Eq. (2) is closely related its underlying topic vector $w$.

---

[1]Strictly speaking, we should first partition $x = (x^{(1)}, x^{(2)})$, only use landmarks that occur as second-halves of documents, and embed $x \mapsto (g^\star(x^{(1)}, l_1), \ldots, g^\star(x^{(1)}, l_M))$. For the sake of clarity, we will ignore this small technical issue here and in the remainder of the paper.

## 4.1 THE SINGLE TOPIC CASE

To build intuition for the embedding in Eq. (2), we first consider the case where each document's probability vector $w$ is supported on a single topic, i.e., $w \in \{e_1, \ldots, e_K\}$ where $e_i$ is the $i^{\text{th}}$ standard basis element. Then we have the following lemma.

**Lemma 1.** *For any documents $x, x'$, we can write $g^\star(x, x') = \eta(x)^\top \psi(x')$, where $\eta(x)_k := \mathbb{P}(w = e_k | x^{(1)} = x)$ is the topic posterior distribution and $\psi(x)_k := \mathbb{P}(x^{(2)} = x | w = e_k)/\mathbb{P}(x^{(2)} = x')$.*

Due to space constraints, all proofs are deferred to Appendix C and Appendix D.

The characterization from Lemma 1 shows that $g^\star$ contains information about the posterior topic distribution $\eta(\cdot)$. To recover it, we must make sure that the $\psi(\cdot)$ vectors for our landmark documents span $\mathbb{R}^K$. Formally, if $l_1, \ldots, l_M$ are the landmarks, and we define the matrix $L \in \mathbb{R}^{K \times M}$ by

$$L := \begin{bmatrix} \psi(l_1) & \cdots & \psi(l_M) \end{bmatrix}, \tag{3}$$

then our representation satisfies $\phi^\star(x) = L^\top \eta(x)$. If our landmarks are chosen so that $L$ has rank $K$, then $L^\dagger \phi^\star(x) = \eta(x)$, where $\dagger$ denotes the matrix pseudo-inverse. Thus, there is a linear transformation of $\phi^\star(x)$ that recovers the posterior distribution of $w$ given $x$.

There are two observations to be made here. The first is that this argument naturally generalizes beyond the single topic setting to any setting where $w$ can take values in a finite set $S$, which may include some mixtures of multiple topics, though the number of landmarks needed would grow at least linearly with $|S|$. The second is that we have made no use of the structure of $x^{(1)}$ and $x^{(2)}$, except for that they are independent conditioned on $w$. Thus, this argument applies to more exotic ways of partitioning a document beyond the bag-of-words approach.

## 4.2 THE GENERAL SETTING

In the general setting, we allow document vectors to be any probability vector in $\Delta^K$, and we do not hope to recover the full posterior distribution over $\Delta^K$. However, the intuition from the single topic case largely carries over, and we will show that we can still recover the posterior moments.

Let $m_{\max}$ be the length of the longest landmark document. Let $S_m^K := \{\alpha \in \mathbb{Z}_+^K : \sum_k \alpha_k = m\}$ denote the set of non-negative integer vectors that sum to $m$ and let $S_{\leq m_{\max}}^K = S_0^K \cup \cdots \cup S_{m_{\max}}^K$. Let $\pi(w)$ denote the degree-$m_{\max}$ monomial vector in $w$: $\pi(w) := (w_1^{\alpha_1} \cdots w_k^{\alpha_k} : \alpha \in S_{\leq m_{\max}}^K)$. For a positive integer $m$ and a vector $\alpha \in S_m^K$, we let $\binom{[m]}{\alpha} := \{z \in [K]^m : \sum_{i=1}^m \mathbb{1}[z_i = k] = \alpha_k \quad \forall k \in [K]\}$. For a document $x$ of length $m$, the degree-$m$ polynomial vector $\psi_m$ is defined by

$$\psi_m(x) := \left( \sum_{z \in \binom{[m]}{\alpha}} \prod_{i=1}^m O(x_i | z_i) : \alpha \in S_m^K \right),$$

and let $\psi_d(x) = \vec{0}$ for all $d \neq m$. The cumulative polynomial vector $\psi$ is given by

$$\psi(x) := \frac{1}{\mathbb{P}(x^{(2)} = x)} (\psi_0(x), \psi_1(x), \cdots, \psi_{m_{\max}}(x)). \tag{4}$$

Given these definitions, we have the following general case analogue of Lemma 1.

**Lemma 2.** *For any documents $x, x'$, we may write $g^\star(x, x') = \eta(x)^\top \psi(x')$ where $\eta(x) := \mathbb{E}[\pi(w) | x^{(1)} = x]$ and $\psi$ is defined in Eq (4).*

Thus, we again have $\phi^\star(x) = L^\top \eta(x)$, but the columns of $L$ are now vectors $\psi(l_i)$ from Eq. (4).

Our analysis, so far, shows that if we choose the landmarks such that $LL^\top$ is invertible, then our representation captures all moments of the topic posterior up to degree $m_{\max}$. As the next theorem shows, we can ensure that $LL^\top$ is invertible whenever each topic has an associated *anchor word* (Arora et al., 2012), i.e., a word that occurs with positive probability only within that topic. In this case, there is a set of landmarks $l_1, \ldots, l_M$ such that any polynomial of $\eta(x)$ can be expressed as a linear function of $\phi^\star(x)$.

**Theorem 3.** *Suppose that (i) each topic has an associated anchor word, and (ii) the marginal distribution of $w$ has positive probability on the interior of $\Delta^K$. For any $d_o \geq 1$, there is a collection of $M = O(K^{d_o})$ landmark documents $l_1, \ldots, l_M$ such that if $Q(w)$ is a degree-$d_o$ polynomial in $w$, then there is a vector $\theta \in \mathbb{R}^M$ such that $\langle \theta, \phi^\star(x) \rangle = \mathbb{E}[Q(w)|x^{(1)} = x]$ for all documents $x$.*

Coupling Theorem 3 with the Stone-Weierstrass theorem (Stone, 1948) shows that, in principle, the posterior mean of any continuous function of $w$ can be approximated using our representation.

## 5 ERROR ANALYSIS

Given a finite amount of data, we cannot hope to solve Eq. (1) exactly. Thus, our solution $\hat{f}$ will only be an approximation to $f^\star$. Since $\hat{f}$ is the basis of our representation, one may worry that errors incurred in this approximation will cascade and cause the approximate representation $\phi(x)$ to differ so wildly from $\phi^\star(x)$ that the results of Section 4 do not even approximately hold.

In this section, we will show that, under certain conditions, such fears are unfounded. Specifically, we will show that there is an error transformation from the approximation error of $\hat{f}$ to the approximation error of linear functions in $\hat{\phi}$. That is, if the target function is $\eta(x)^\mathsf{T}\theta^\star$, then we will show that the best mean squared error achievable using our approximate representation $\hat{\phi}$, given by

$$R(\hat{\phi}) := \min_v \mathbb{E}_{x \sim \mu^{(1)}} (\eta(x)^\mathsf{T}\theta^\star - \hat{\phi}(x)^\mathsf{T}v)^2,$$

is bounded in terms of the approximation quality of $\hat{f}$ as well as some other terms. Here, $\mu^{(1)}$ is the marginal distribution over first halves of documents drawn from $\mathcal{D}$. Thus, for the specific setting of semi-supervised learning, an approximate solution to Eq. (1) is good enough.

There are a number of reasonable ways to choose landmark documents. Here we consider a simple method: randomly sample them from the marginal distribution $\mu^{(2)}$ of $x^{(2)}$. We will assume that this distribution satisfies certain regularity properties.

**Assumption 1.** *There is a constant $\sigma_{\min} > 0$ such that for any $\delta \in (0, 1)$, there is a number $M_0$ such that for an i.i.d. sample $l_1, \ldots, l_M$ from $\mu^{(2)}$, with $M \geq M_0$, with probability $1 - \delta$, the matrix $L$ in Eq. (3) (with $\psi$ as defined in Lemma 1 or Eq. (4)) has minimum singular value at least $\sigma_{\min}\sqrt{M}$.*

Note that the smallest non-zero singular value of $\frac{1}{\sqrt{M}}L$ is the square-root of the smallest eigenvalue of a certain empirical second-moment matrix. Hence, Assumption 1 holds under appropriate conditions on the landmark distribution, for instance via tail bounds for sums of random matrices (Tropp, 2012) combined with matrix perturbation analysis (e.g., Weyl's inequality). In the single topic setting with anchor words, it can be shown that for long enough documents, $\sigma_{\min}$ is lower-bounded by a constant when $M_0$ grows polynomially with $K$. We defer a detailed proof of this to Appendix D.

We will also assume that the predictions of $\hat{f}$ and $f^\star$ bounded above by some constant.

**Assumption 2.** *There exists some $g_{\max} > 0$ such that $\hat{f}(x, l_i), f^\star(x, l_i) \leq \log g_{\max}$ for all documents $x$ and landmarks $l_i$.*

Note that if Assumption 2 holds for $f^\star$, then it can be made to hold for $\hat{f}$ by clipping. Moreover, it holds for $f^\star$ whenever the vocabulary and document sizes are constants:

$$f^\star(x, x') = \log \frac{\mathbb{P}(x^{(1)} = x, x^{(2)} = x')}{\mathbb{P}(x^{(1)} = x)\mathbb{P}(x^{(2)} = x')} = \log \frac{\mathbb{P}(x^{(2)} = x' \mid x^{(1)} = x)}{\mathbb{P}(x^{(2)} = x')} \leq \log \frac{1}{\mathbb{P}(x^{(2)} = x')}.$$

Since landmarks are sampled, and the number of possible documents is finite, there exists a constant $p_{\min} > 0$ such that $\mathbb{P}(x^{(2)} = l) \geq p_{\min}$. Thus, Assumption 2 holds for $g_{\max} \leq 1/p_{\min}$.

Given these assumptions, we have the following error transformation guarantee.

**Theorem 4.** *Fix any $\delta \in (0, 1)$, and suppose Assumption 1 and Assumption 2 hold (with $M_0$, $\sigma_{\min}$, and $f_{\max}$). Let $\hat{f}$ be the function returned by the contrastive learning algorithm, and let $\varepsilon := L_{\text{contrast}}(\hat{f}) - L_{\text{contrast}}(f^\star)$ denote its excess contrastive loss. If $M \geq M_0$, then with probability*

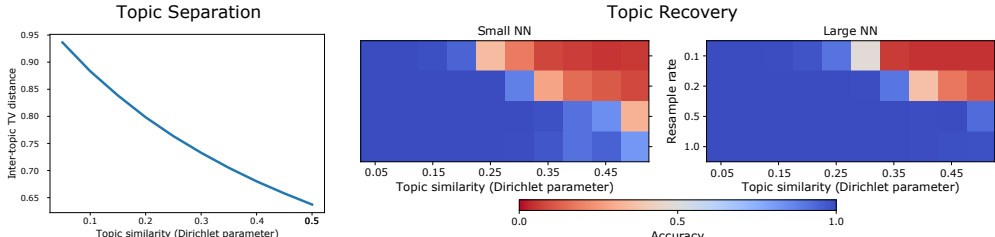

Figure 1: Topic modeling simulations. Left: Average total variation distance between topics. Right: Topic recovery accuracy for contrastive models. Total number of documents sampled $= 6M \times$ rate.

*at least $1 - \delta$ over the random sample of $l_1, \ldots l_M$,*

$$R(\hat{\phi}) \leq \frac{\|\theta^\star\|_2^2 (1 + g_{\max})^4}{\sigma_{\min}^2} \left( \varepsilon + \sqrt{\frac{2 \log(2/\delta)}{M}} \right).$$

We make a few observations here. First, $\|\theta^\star\|_2^2$ is a measure of the complexity of the target function. Thus, if the target function is some reasonable function (e.g., low-degree polynomial) of the posterior document vector, then we can expect $\|\theta^\star\|_2^2$ to be small. Second, the dependence on $g_{\max}$ is probably not very tight and can likely be improved. Third, note that $M$ can grow and $\varepsilon$ can shrink with the number of *unlabeled* documents; indeed, none of the terms in Theorem 4 deal with labeled data.

Finally, it is possible to establish guarantees in a semi-supervised setting using our analysis. If we have $n_L$ i.i.d. labeled examples, and we learn a linear predictor $\hat{v}$ with the representation $\hat{\phi}$ using ERM (say), then the bias-variance decomposition grants

$$\mathrm{mse}(\hat{v}) := \mathbb{E}_{x \sim \mu^{(1)}} (\eta(x)^\mathsf{T} \theta^\star - \hat{\phi}(x)^\mathsf{T} \hat{v})^2 = R(\hat{\phi}) + \mathbb{E}_{x \sim \mu^{(1)}} (\hat{\phi}(x)^\mathsf{T} (v^* - \hat{v}))^2,$$

where $v^*$ is the minimizer of $\mathrm{mse}(\cdot)$. The final term $\mathbb{E}_{x \sim \mu^{(1)}} (\hat{\phi}(x)^\mathsf{T} (v^* - \hat{v}))^2$ is the excess risk in linear regression, which goes to zero as $n_L \to \infty$.

## 6 TOPIC MODELING SIMULATIONS

To test our theory, we ran simulation experiments with a single-topic generative model where $K = 20$ topics are sampled from a symmetric Dirichlet($\alpha$) distribution over a vocabulary of size 5k. The Dirichlet parameter $\alpha$ governs the sparsity of the topic distributions, effectively determining the similarity of the topics: as $\alpha$ increases the prior concentrates on the interior of the simplex, forcing the topic distributions to be more similar. This is visualized in the left panel of Figure 1.

In the experiments, we generate a dataset and solve the contrastive optimization problem, and then we construct the landmark embeddings $\phi(x)$ for each document $x$ using 1k landmark documents, following Section 4. Using the true likelihood matrix $L$ for the landmarks, we infer the MAP topic estimate and measure accuracy as the fraction of test documents for which this prediction matches the generating topic. See Appendix A for additional details.

The results are displayed in the center and right panel of Figure 1 where we vary the network architecture and the amount of training data. The experiment identifies several interesting properties of the contrastive learning approach. First, as a sanity check, the algorithm does accurately predict the latent topics of the test documents in most experimental conditions and the accuracy is quite high when the problem is relatively easy (e.g., $\alpha$ is small). Second, the performance degrades as $\alpha$ increases, but this can be mitigated by increasing the model capacity (size of the network) or the resampling rate (which exposes the model to more unlabeled data). Specifically, we consistently see that for a fixed model and $\alpha$, increasing the resampling rate improves accuracy. A similar trend emerges when we fix $\alpha$ and rate and increase model capacity. These findings suggest that latent topics can be recovered by the contrastive learning approach, provided we have an expressive enough model and enough unlabeled data.

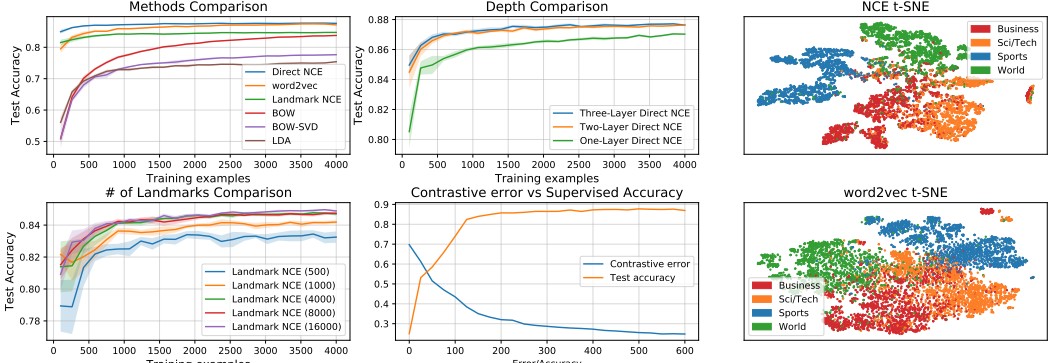

Figure 2: Experiments with AG news dataset. Top left: test accuracy of methods as we increase the number of supervised training examples. Bottom left: `Landmark-NCE` performance as we vary number of landmarks. Top middle: `Direct-NCE` performance as we vary network depth. Bottom middle: Relationship between contrastive error and test accuracy for `Direct-NCE`. Right: t-SNE visualizations of `Direct-NCE` and `word2vec` embeddings.

## 7 SEMI-SUPERVISED EXPERIMENTS

We also conducted experiments with our document-level contrastive representations in a semi-supervised setting. The goal of these experiments is to demonstrate that the contrastive representations yield non-trivial performance, as consistent with the theory. Note that our intention is *not* to show state-of-the-art performance using contrastive learning; that is beyond the scope of the paper.

We discuss the main findings here, with experimental details deferred to Appendix B.

**A closely related representation.** In the worst-case, the guarantees from Section 4 and Section 5 require the number of landmarks to be quite large. To develop a more practical representation, and to more closely mirror what is done in practice, we consider training models of the form $f_1, f_2 : \mathcal{X} \to \mathbb{R}^d$ where $(x, x') \mapsto f_1(x)^\intercal f_2(x')$. Plugging this into Eq (1), we solve the following bivariate optimization problem:

$$\underset{f_1, f_2}{\text{minimize}}\, \mathbb{E}_{\mathcal{D}_{\text{contrast}}} \left[ y \log \left( 1 + \exp \left( -f_1(x)^\intercal f_2(x') \right) \right) + (1 - y) \log \left( 1 + \exp \left( f_1(x)^\intercal f_2(x') \right) \right) \right].$$
(5)

Given $f_1, f_2$, we can embed a document $x$ according to $f_1(x)$. We call the resulting scheme the *direct embedding* approach to distinguish it from the *landmark embedding* approach from Section 3.

**Methodology.** We used the AG news topic classification dataset (Zhang et al., 2015), which has 4 classes and 30k training examples per class. We reserve 1k examples per class as labeled training data and use the remaining examples for representation learning. For all methods, we use $\ell_2$-regularized logistic regression to fit a linear classifier on the labeled data.

We compared the representations `Landmark-NCE` and `Direct-NCE` against the following baselines: (1) standard bag-of-words (`BOW`), (2) bag-of-words with dimensionality reduction (`BOW+SVD`), (3) representations from LDA (`LDA`), and (4) skip-gram word embeddings (Mikolov et al., 2013b) (`word2vec`). For the `NCE` methods, we experiment with different neural network architectures and numbers of landmarks but use standard settings for other training parameters. See Appendix B for details. We note that all of these methods all of these methods ignore word order in the final document-level representation, and all of them (with the exception of `word2vec`) ignore word order in their training.

In all line plots in Figure 2, the training examples axis refers to the number of randomly selected labeled examples used to train the linear classifier. The shaded regions denote 95% confidence intervals computed over 10 replicates of this random selection procedure.

**Baseline comparison.** In the left panel of Figure 2, we visualize the semi-supervised performance of `NCE` and the baselines. `Direct-NCE` outperforms all the other methods, with dramatic improve-

ments over all except `word2vec` in the low labeled data regime. `BOW` is quite competitive when there is an abundance of labeled data, but as the dimensionality of this representation is quite large, it performs poorly with limited samples. However, unsupervised dimensionality reduction on this representation appears to be unhelpful and actually degrades performance uniformly. Finally, we point out that word embedding representations (`word2vec`) perform quite well, but our document-level `Direct-NCE` procedure is slightly better, particularly when there are few labeled examples. This may reflect some advantage in learning document-level non-linear representations, as opposed to averaging word-level ones.

**Visualizing embeddings.** For a qualitative perspective, we visualize the embeddings from `NCE` using t-SNE with the default scikit-learn parameters (van der Maaten & Hinton, 2008; Pedregosa et al., 2011). To compare, we also used t-SNE to visualize the document-averaged `word2vec` embeddings. The right panels of Figure 2 shows these visualizations on the 7,600 test documents colored according to their true label. While qualitiative, the visualization of the `Direct-NCE` embeddings appear to be more clearly separated into label-homogeneous regions than that of `word2vec`.

**Other results.** We investigated the effect of the number of landmarks on the performance of `Landmark-NCE` by embedding with 500, 1k, 4k, 8k, and 16k landmarks. The bottom left panel of Figure 2 displays the results, which suggest that a larger number of landmarks is helpful, with diminishing returns at the higher end of the scale.

We also looked into the effect of depth on the performance of `Direct-NCE` by training networks with one, two, and three hidden layers. In each case, the first hidden layer has 300 nodes and the rest have 256 nodes. The top center panel of Figure 2 displays the results, which suggest that using deeper models for representation learning may improve downstream performance.

We also tracked the contrastive loss of the model on a holdout validation contrastive dataset. The bottom center panel of Figure 2 plots how this loss evolves over training epochs. Along with this contrastive loss, we checkpoint the model, train a linear classifier, and evaluate the supervised test accuracy. We see that test accuracy steadily improves as contrastive loss decreases, suggesting that in these settings, contrastive loss (which we can measure using an unlabeled validation set) is a good surrogate for downstream performance (which may not be measurable until we have a task at hand).

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

## A    Topic modeling simulations

The results of Section 4 show that if a model is trained to minimize the contrastive learning objective, then that model must also recover certain topic posterior information in the corpus. However, there are a few practical questions that remain: can we train such a model, how much capacity should it have, and how much data is needed in order to train it? Our topic modeling simulations are designed to study these questions.

**Simulation setup.**    We considered a single-topic generative model where the topics $\theta_1, \ldots, \theta_K$ are sampled from a symmetric Dirichlet$(\alpha/K)$ distribution over $\Delta^{|\mathcal{V}|}$ and for each document, its length is drawn from a Poisson$(\lambda)$ and its topic is sampled uniformly from $[K]$. This model can be thought of as a limiting case of the LDA model (Blei et al., 2003; Griffiths & Steyvers, 2004) when the document-level topic distribution is symmetric Dirichlet$(\beta)$ with $\beta \ll 1$. In our experiments, we set $K = 20$, $|\mathcal{V}| = 5000$, and $\lambda = 30$, and we varied $\alpha$ from 1 to 10. Notice that as $\alpha$ increases, the Dirichlet prior becomes more concentrated around the uniform distribution, so the topic distributions are more likely to be similar. Thus, we expect the contrastive learning problem to be more difficult with larger values of $\alpha$.

We used contrastive models of the same form as Section 7, namely models of the form $f_1, f_2$ where the final prediction is $f_1(x)^\intercal f_2(x')$ and $f_1$ and $f_2$ are fully-connected neural networks with three hidden layers. To measure the effect of model capacity, we trained two models – a smaller model with 256 nodes per hidden layer and a larger model with 512 nodes per hidden layer. Both models were trained for 100 epochs. We used all of the same optimization parameters as in Section 7 with the exception of dropout, which we did not use.

To study the effect of training data, we varied the rate $r$ at which we resampled our entire contrastive training set from the ground truth topic model. Specifically, after every $1/r$-th training epoch, we resampled 60,000 new documents and constructed a contrastive dataset from these documents. We varied the resampling rate $r$ from 0.1 to 1.0, where larger values of $r$ imply more training data. The total amount of training data varies from 600K documents to 6M documents.

Using the results from Section 4, we constructed the embedding $\phi(x)$ of a new document $x$ using 1000 landmark documents, each sampled from the same generative model. We constructed the true likelihood matrix $L$ of the landmark documents using the underlying topic model and recovered the model-based posterior $L^\dagger \phi(x)$. We measured accuracy as the fraction of testing documents for which the MAP topic under the model-based posterior matched the generating topic. We used 5000 testing documents and performed 5 replicates for each setting of parameters.

## B    Semi-supervised experiment details

**Methodology.**    We conducted semi-supervised experiments on the AG news topic classification dataset as compiled by Zhang et al. (2015). This dataset contains news articles that belong to one of four categories: world, sports, business, and sci/tech. There are 30,000 examples from each class in the training set, and 1,900 examples from each class in the testing set. We minimally preprocessed the dataset by removing punctuation and words that occurred in fewer than 10 documents, resulting in a vocabulary of approximately 16,700 words.

We randomly selected 1,000 examples from each class to remain as our labeled training dataset, and we used the remaining 116,000 examples as our unlabeled dataset for learning representations. After computing representations on the unlabeled dataset, we fit a linear classifier on the labeled training set using logistic regression with cross validation to choose the $\ell_2$ regularization parameter ($n_{\text{folds}} = 3$).

We compared our representations, `Landmark-NCE` and `Direct-NCE`, against several representation baselines.

- `BOW` – The standard bag-of-words representation.
- `BOW+SVD` – A bag of words representation with dimensionality reduction. We first perform SVD on the bag-of-words representation using the unsupervised dataset to compute

a low dimensional subspace, and train a linear classifier on the projected bag-of-words representations with the labeled dataset.

- `LDA` – A representation derived from LDA. We fit LDA on the unsupervised dataset using online variational Bayes (Hoffman et al., 2010), and our representation is the inferred posterior distribution over topics given training document.

- `word2vec` – Skip-gram word embeddings (Mikolov et al., 2013b). We fit the skip-gram word embeddings model on the unsupervised dataset and then averaged the word embeddings in each of the training documents to get their representation.

For our representation, to solve the optimization problem in Eq. (5), we considered neural network architectures of various depths. We used fully-connected layers with between 250 and 300 nodes per hidden layer. We used ReLU nonlinearities, dropout probability 1/2, batch normalization, and the default PyTorch initialization (Paszke et al., 2019). We optimized using RMSProp with momentum value 0.009 and weight decay 0.0001 as in Radhakrishnan et al. (2019). We started with learning rate $10^{-4}$ which we halved after 250 epochs, and we trained for 600 epochs. Unless otherwise stated, `Landmark-NCE` and `Direct-NCE` use a three-layer architecture, and `Landmark-NCE` uses 8000 landmarks.

To sample a contrastive dataset, we first randomly partitioned each unlabeled document in half to create the positive pairs. To create the negative pairs, we again randomly partitioned each unlabeled document in half, randomly permuted one set of half documents, and discarded collisions. This results in a contrastive dataset whose size is roughly twice the number of unlabeled documents. In the course of training our models for the contrastive task, we resampled a contrastive dataset every 3 epochs to prevent overfitting on any one particular dataset.

**Additional discussions.** In the left panel of Figure 2, we additionally remark that `LDA` performs quite poorly. This could be for several reasons, including that fitting a topic model directly could be challenging on the relatively short documents in the corpus or that the document category is not well-expressed by a linear function of the topic proportions.

## C  PROOFS FROM SECTION 4

### C.1  PROOF OF SINGLE TOPIC REPRESENTATION LEMMA

*Proof of Lemma 1.* Conditioned on the topic vector $w$, $x^{(1)}$ and $x^{(2)}$ are independent. Thus,

$$
\begin{aligned}
g^\star(x, x') &= \frac{\mathbb{P}(x^{(1)} = x, x^{(2)} = x')}{\mathbb{P}(x^{(1)} = x)\mathbb{P}(x^{(2)} = x')} \\
&= \sum_{k=1}^{K} \frac{\mathbb{P}(w = e_k)\mathbb{P}(x^{(1)} = x \mid w = e_k)\mathbb{P}(x^{(2)} = x' \mid w = e_k)}{\mathbb{P}(x^{(1)} = x)\mathbb{P}(x^{(2)} = x')} \\
&= \sum_{k=1}^{K} \frac{\mathbb{P}(w = e_k \mid x^{(1)} = x)\mathbb{P}(x^{(2)} = x' \mid w = e_k)}{\mathbb{P}(x^{(2)} = x')} \\
&= \frac{\eta(x)^\top \psi(x')}{\mathbb{P}(x^{(2)} = x')},
\end{aligned}
$$

where the third equality follows from Bayes' rule. □

### C.2  PROOF OF GENERAL REPRESENTATION LEMMA

*Proof of Lemma 2.* Fix a document $x$ of length $m$ and a document probability vector $w$. Conditioned on the assignment of each word in the document to a topic, probability of a document factorizes as

$$
\mathbb{P}(x \mid w) = \sum_{z \in [K]^m} \prod_{i=1}^{m} w_{z_i} O(x_i \mid z_i) = \sum_{z \in [K]^m} \left(\prod_{i=1}^{m} w_{z_i}\right) \left(\prod_{i=1}^{m} O(x_i \mid z_i)\right) = \pi(w)^\top \psi(x),
$$

where the last line follows from collecting like terms. Using the form of $g^\star$ from above, we have

$$
\begin{aligned}
g^\star(x, x') &= \frac{\mathbb{P}(x^{(1)} = x, x^{(2)} = x')}{\mathbb{P}(x^{(1)} = x)\mathbb{P}(x^{(2)} = x')} \\
&= \frac{\int_w \mathbb{P}(x^{(1)} = x \mid w)\mathbb{P}(x^{(2)} = x' \mid w) \, \mathrm{d}\mathbb{P}(w)}{\mathbb{P}(x^{(1)} = x)\mathbb{P}(x^{(2)} = x')} \\
&= \frac{\int_w \mathbb{P}(x^{(2)} = x' \mid w) \, \mathrm{d}\mathbb{P}(w \mid x^{(1)} = x)}{\mathbb{P}(x^{(2)} = x')} \\
&= \frac{\int_w \pi(w)^\intercal \psi(x) \, \mathrm{d}\mathbb{P}(w \mid x^{(1)} = x)}{\mathbb{P}(x^{(2)} = x')} \\
&= \frac{\eta(x)^\intercal \psi(x')}{\mathbb{P}(x^{(2)} = x')}. \qquad \qquad \square
\end{aligned}
$$

### C.3 PROOF OF POLYNOMIAL REPRESENTATION THEOREM

*Proof of Theorem 3.* By assumption (i), there exists an anchor word $a_k$ for each topic $k = 1, \ldots, K$. By definition this means that $O(a_k \mid j) > 0$ if and only if $j = k$. For each vector $\alpha \in \mathbb{Z}_+^K$ such that $\sum \alpha_k \leq d_o$, create a landmark document consisting of $\alpha_k$ copies of $a_k$ for $k = 1, \ldots, K$. This will result in $\binom{K+d_o}{d_o}$ landmark documents. Moreover, from assumption (ii), we can see that each of these landmark documents has positive probability of occurring under the marginal distribution of $x^{(2)}$ for $(x^{(1)}, x^{(2)}, y) \sim \mathcal{D}_{\text{contrast}}$, which implies $g^\star(x, l)$ is well-defined for all our landmark documents $l$.

Let $l$ denote one of our landmark documents and let $\alpha \in \mathbb{Z}_+^K$ be its associated vector. Since $l$ only contains anchor words, $\psi(l)_\beta > 0$ if and only if $\alpha = \beta$. To see this, note that

$$
\psi(l)_\alpha = \sum_{z \in \binom{[m]}{\alpha}} \prod_{i=1}^m O(l_i \mid z_i) \geq \prod_{k=1}^K O(a_k \mid k)^{\alpha_k} > 0.
$$

On the other hand, if $\beta \neq \alpha$ but $\sum_k \beta_k = \sum_k \alpha_k$, then there exists an index $k$ such that $\beta_k \geq \alpha_k + 1$. Thus, for any $z \in \binom{[m]}{\beta}$, there will be more than $\alpha_k$ words in $l$ assigned to topic $k$. Since every word in $l$ is an anchor word and at most $\alpha_k$ of them correspond to topic $k$, we will have

$$
\prod_{i=1}^m O(l_i \mid z_i) = 0.
$$

Rebinding $\psi(l) = (\psi_0(l), \ldots, \psi_{d_0}(l))$ and forming the matrix $L$ using this definition, we see that $L^\intercal$ can be diagonalized and inverted.

For any target degree-$d_o$ polynomial $Q(w)$, there exists a vector $v$ such that $Q(w) = \langle v, \pi_{d_0}(w) \rangle$, where $\pi_{d_0}(w)$ denotes the degree-$d_0$ monomial vector. Thus, we may take $\theta = L^{-1}v$ and get that for any document $x$:

$$
\langle \theta, g^\star(x, l_{1:M}) \rangle = (L^{-1}v)^T L^\intercal \eta(x) = \mathbb{E}[\langle v, \pi_{d_0}(w) \rangle \mid x^{(1)} = x] = \mathbb{E}[Q(w) \mid x^{(1)} = x]. \qquad \square
$$

## D PROOFS FROM SECTION 5

### D.1 PROOF OF ERROR TRANSFORMATION GUARANTEE

We first recall and setup some notations. For $(x^{(1)}, x^{(2)}, y) \sim \mathcal{D}_{\text{contrast}}$ (our contrastive distribution defined in Section 3), we let $\mu_i$ denote the marginal distribution of $x^{(i)}$. Furthermore, recall the

contrastive loss, conditional probability, odds ratio, and oracle representation functions:

$$L_{\text{contrast}}(f) := \mathbb{E}_{(x,x',y)\sim\mathcal{D}_{\text{contrast}}} \left[ y \log \left( 1 + e^{-f(x,x')} \right) + (1 - y) \log \left( 1 + e^{f(x,x')} \right) \right]$$

$$f^\star(x, x') := \log \frac{\mathbb{P}(y = 1 \mid x^{(1)} = x, x^{(2)} = x')}{\mathbb{P}(y = 0 \mid x^{(1)} = x, x^{(2)} = x')},$$

$$g^\star(x, x') := \exp \left( f^\star(x, x') \right) = \frac{\mathbb{P}(x^{(1)} = x, x^{(2)} = x')}{\mathbb{P}(x^{(1)} = x)\mathbb{P}(x^{(2)} = x')},$$

$$\phi^\star(x) := (g^\star(x, l_1), \ldots, g^\star(x, l_M))$$

where $l_1, \ldots, l_M$ are landmark documents. The learned approximation to $f^\star$ is $\hat{f}$, and from it we derive

$$\hat{g}(x, x') := \exp \left( \hat{f}(x, x') \right),$$

$$\hat{\phi}(x) := (\hat{g}(x, l_1), \ldots, \hat{g}(x, l_M))$$

Let $\eta(x), \psi(x)$ denote the posterior/likelihood vectors from Lemma 1 or the posterior/likelihood polynomial vectors from Lemma 2. Say the length of this vector is $N \geq 1$.

Our goal is to show that linear functions in the representation $\hat{\phi}(x)$ can provide a good approximation to the target function

$$x \mapsto \eta(x)^\mathsf{T}\theta^\star$$

where $\theta^\star \in \mathbb{R}^N$ is some fixed vector. To this end, define

$$R(\hat{\phi}) := \min_v \mathbb{E}_{x\sim\mu^{(1)}}(\eta(x)^\mathsf{T}\theta^\star - \hat{\phi}(x)^\mathsf{T}v)^2,$$

which is the best mean squared error achievable using the representation $\hat{\phi}$.

By Lemma 1 or Lemma 2, we know that for any $x, x'$ we have

$$g^\star(x, x') = \eta(x)^\mathsf{T}\psi(x').$$

Recall the matrix

$$L := \left( \psi(l_1), \ldots, \psi(l_M) \right).$$

This matrix is in $\mathbb{R}^{N \times M}$. If $L$ has full row rank, then

$$\eta(x)^\mathsf{T}\theta^\star = \eta(x)^\mathsf{T}LL^\dagger\theta^\star = \phi^\star(x)^\mathsf{T}v^\star$$

where

$$\phi^\star(x) := (g^\star(x, l_1), \ldots, g^\star(x, l_M))$$

and $v^\star = L^\dagger\theta^\star$. Thus, $R(\phi^\star) = 0$. We will show that $R(\hat{\phi})$ can be bounded as well.

**Theorem 5** (Restatement of Theorem 4). *Suppose the following assumptions hold.*

*(1) There is a constant $\sigma_{\min} > 0$ such that for any $\delta \in (0, 1)$, there is a number $M_0(\delta)$ such that for an i.i.d. sample $l_1, \ldots, l_M$ with $M \geq M_0(\delta)$, with probability $1 - \delta$, the matrix*

$$L = \begin{bmatrix} \psi(l_1) & \cdots & \psi(l_M) \end{bmatrix}$$

*has minimum singular value at least $\sigma_{\min}\sqrt{M}$.*

*(2) There exists a value $g_{\max} > 0$ such that for all documents $x$ and landmarks $l_i$*

$$\max\{\hat{f}(x, l_i), f^\star(x, l_i)\} \leq \log g_{\max}.$$

*Let $\hat{f}$ be the function returned by the contrastive learning algorithm, and let*

$$\varepsilon := L_{\text{contrast}}(\hat{f}) - L_{\text{contrast}}(f^\star)$$

*denote its excess contrastive loss. For any $\delta \in (0, 1)$, if $M \geq M_0(\delta/2)$, then with probability at least $1 - \delta$ over the random draw of $l_1, \ldots, l_M$, we have*

$$R(\hat{\phi}) \leq \frac{\|\theta^\star\|_2^2 (1 + g_{\max})^4}{\sigma_{\min}^2} \left( \varepsilon + \sqrt{\frac{2 \log(2/\delta)}{M}} \right).$$

*Proof.* We first condition on two events based on the sample $l_1, \ldots, l_M$. The first is the event that $L$ has full row rank and smallest non-zero singular value at least $\sqrt{M}\sigma_{\min} > 0$; this event has probability at least $1 - \delta/2$. The second is the event that

$$\frac{1}{M} \sum_{j=1}^{M} \mathbb{E}_{x \sim \mu^{(1)}} \left(p^\star(x, x') - \hat{p}(x, x')\right)^2 \leq \mathbb{E}_{(x,x') \sim \mu^{(1)} \otimes \mu^{(2)}} \left(p^\star(x, l_j) - \hat{p}(x, l_j)\right)^2 + \sqrt{\frac{2 \log(2/\delta)}{M}} \tag{6}$$

where we make the definitions

$$\hat{g}(x, x') := \exp(\hat{f}(x, x'))$$

$$\hat{p}(x, x') := 1/(1 + e^{-\hat{f}(x,x')}) = \frac{\hat{g}(x, x')}{1 + \hat{g}(x, x')}$$

$$p^\star(x, x') := 1/(1 + e^{-f^\star(x,x')}) = \frac{g^\star(x, x')}{1 + g^\star(x, x')}.$$

By Hoeffding's inequality and the fact that $\hat{p}$ and $p^\star$ have range $[0, 1]$, this event also has probability at least $1 - \delta/2$. By the union bound, both events hold simultaneously with probability at least $1 - \delta$. We henceforth condition on these two events for the remainder of the proof.

Since $L$ has full row rank, via Cauchy-Schwarz, we have

$$R(\hat{\phi}) = \min_v \mathbb{E}_{x \sim \mu^{(1)}} (\eta(x)^\top \theta^\star - \hat{\phi}(x)^\top v)^2$$

$$\leq \mathbb{E}_{x \sim \mu^{(1)}} (\eta(x)^\top \theta^\star - \hat{\phi}(x)^\top v^\star)^2$$

$$= \mathbb{E}_{x \sim \mu^{(1)}} ((\phi^\star(x)^\top - \hat{\phi}(x))^\top v^\star)^2$$

$$\leq \mathbb{E}_{x \sim \mu^{(1)}} \|v^\star\|_2^2 \left\|\phi^\star(x)^\top - \hat{\phi}(x)\right\|_2^2$$

$$= \|v^\star\|_2^2 \cdot \mathbb{E}_{x \sim \mu^{(1)}} \left\|\phi^\star(x)^\top - \hat{\phi}(x)\right\|_2^2.$$

We analyze the two factors on the right-hand side separately.

**Analysis of $v^\star$.** For $v^\star$, we have

$$\|v^\star\|_2^2 \leq \left\|L^\dagger\right\|_2^2 \|\theta^\star\|_2^2 \leq \frac{1}{M\sigma_{\min}^2} \|\theta^\star\|_2^2,$$

where we have used the fact that $L$ has smallest non-zero singular value at least $\sqrt{M}\sigma_{\min}$.

**Analysis of $\phi^\star - \hat{\phi}$.** For the other term, we first note that

$$p^\star(x, x') = 1/(1 + e^{-f^\star(x,x')}) = \mathbb{P}(y = 1 \mid x^{(1)} = x, x^{(2)} = x').$$

Thus, we have

$$\varepsilon = L_{\text{contrast}}(\hat{f}) - L_{\text{contrast}}(f^\star)$$

$$= \mathbb{E}_{(x,x',y) \sim \mathcal{D}_{\text{contrast}}} \left[y \log\left(\frac{p^\star(x, x')}{\hat{p}(x, x')}\right) + (1 - y) \log\left(\frac{1 - p^\star(x, x')}{1 - \hat{p}(x, x')}\right)\right]$$

$$= \mathbb{E}_{(x,x') \sim \mathcal{D}_{\text{contrast}}} \left[p^\star(x, x') \log\left(\frac{p^\star(x, x')}{\hat{p}(x, x')}\right) + (1 - p^\star(x, x')) \log\left(\frac{1 - p^\star(x, x')}{1 - \hat{p}(x, x')}\right)\right]$$

$$= \mathbb{E}_{(x,x') \sim \mathcal{D}_{\text{contrast}}} \left[\text{KL}(p(x, x'), p^\star(x, x'))\right]$$

where $\text{KL}(p, p')$ denotes the KL-divergence between two Bernoulli distributions with biases $p$ and $p'$, respectively. Pinsker's inequality tells us that $\text{KL}(p, p') \geq 2(p - p')^2$. Combining this with the fact that $\mathcal{D}_{\text{contrast}}$ is a mixture distribution that places half its probability mass in $\mu^{(1)} \otimes \mu^{(2)}$ implies

$$\varepsilon \geq 2\mathbb{E}_{(x,x') \sim \mathcal{D}_{\text{contrast}}} \left[\left(\hat{p}(x, x') - p^\star(x, x')\right)^2\right] \mathbb{E}_{(x,x') \sim \mu^{(1)} \otimes \mu^{(2)}} \left[\left(\hat{p}(x, x') - p^\star(x, x')\right)^2\right].$$

Combining the above with Eq. (6) and the definitions of $\hat{p}, p^\star$, we have

$$
\begin{aligned}
\mathbb{E}_{x \sim \mu^{(1)}} \left\| \phi^\star(x) - \hat{\phi}(x) \right\|_2^2 &= \sum_{j=1}^M \mathbb{E}_{x \sim \mu^{(1)}} (g^\star(x, l_j) - \hat{g}(x, l_j))^2 \\
&\leq (1 + g_{\max})^4 \sum_{j=1}^M \mathbb{E}_{x \sim \mu^{(1)}} (p^\star(x, l_j) - \hat{p}(x, l_j))^2 \\
&\leq M(1 + g_{\max})^4 \left( \mathbb{E}_{(x,x') \sim \mu^{(1)} \otimes \mu^{(2)}} \left( p^\star(x, l_j) - \hat{p}(x, l_j) \right)^2 + \sqrt{\frac{2 \log(2/\delta)}{M}} \right) \\
&\leq M(1 + g_{\max})^4 \left( \varepsilon + \sqrt{\frac{2 \log(2/\delta)}{M}} \right).
\end{aligned}
$$

**Wrapping up.** To conclude, we have

$$
\begin{aligned}
R(\hat{\phi}) &\leq \|v^\star\|_2^2 \cdot \mathbb{E}_{x \sim \mu^{(1)}} \left\| \phi^\star(x)^\mathsf{T} - \hat{\phi}(x) \right\|_2^2 \\
&\leq \left( \frac{1}{M \sigma_{\min}^2} \|\theta^\star\|_2^2 \right) \left( M(1 + g_{\max})^4 \left( \varepsilon + \sqrt{\frac{2 \log(2/\delta)}{M}} \right) \right) \\
&= \frac{\|\theta^\star\|_2^2 (1 + g_{\max})^4}{\sigma_{\min}^2} \left( \varepsilon + \sqrt{\frac{2 \log(2/\delta)}{M}} \right). \qquad \square
\end{aligned}
$$

### D.2 SATISFYING ASSUMPTION 1

Suppose we are in the single topic case where $w \in \{e_1, \ldots, e_K\}$. Assume that $\min_k \Pr(w = e_k) \geq w_{\min}$. Further assumes that each topic $k$ has an anchor word $a_k$, satisfying $O(a_k \mid z = e_k) \geq a_{\min}$. Then we will show that when $M$ and $m$ are large enough, the matrix $L$ whose columns are $\psi(x)/\mathbb{P}(x)$ will have large singular values.

First note that if document $x$ contains $a_k$ then $\psi(x)$ is one sparse, and satisfies

$$
\text{if } a_k \in x: \qquad \psi(x) = \frac{\mathbb{P}(x \mid w = e_k)}{\sum_{k'} \mathbb{P}(w = k') \mathbb{P}(x \mid w = k')} e_k = \frac{1}{\mathbb{P}(w = k')} e_k.
$$

Therefore, the second moment matrix satisfies

$$
\begin{aligned}
\mathbb{E} \left[ \psi(x) \psi(x)^\mathsf{T} \right] &\succeq \sum_{k=1}^K \mathbb{P}(w = e_k) \mathbb{P}(a_k \in x \mid e_k) \mathbb{E} \left[ \psi(x) \psi(x)^\mathsf{T} \mid a_k \in x, w = e_k \right] \\
&= \sum_{k=1}^K \frac{\mathbb{P}(a_k \in x \mid e_k)}{\mathbb{P}(w = e_k)} e_k e_k^\mathsf{T}.
\end{aligned}
$$

Now, if the number of words per document is $m \geq 1/a_{\min}$ then

$$
\begin{aligned}
\mathbb{P}(a_k \in x \mid e_k) = 1 - (1 - O(a_k \mid e_k))^m &\geq 1 - \exp(-m O(a_k \mid e_k)) \\
&\geq 1 - \exp(-m a_{\min}) \\
&\geq 1 - 1/e.
\end{aligned}
$$

Finally, using the fact that $\mathbb{P}(w = e_k) \leq 1$, we see that the second moment matrix satisfies

$$
\mathbb{E} \left[ \psi(x) \psi(x)^\mathsf{T} \right] \succeq (1 - 1/e) I_{K \times K}.
$$

For the empirical matrix, we perform a crude analysis and apply the Matrix-Hoeffding inequality (Tropp, 2012). We have $\left\| \psi(x) \psi(x)^\mathsf{T} \right\|_2 \leq K w_{\min}^{-2}$ and so with probability at least $1 - \delta$, we have

$$
\left\| \frac{1}{M} \sum_{i=1}^M \psi(l_i) \psi(l_i)^\mathsf{T} - \mathbb{E} \left[ \psi(x) \psi(x)^\mathsf{T} \right] \right\|_2 \leq \sqrt{\frac{8K \log(K/\delta)}{M w_{\min}^2}}.
$$

If we take $M \geq \Omega(K \log(K/\delta)/w_{\min}^2)$ then we will have that the minimum eigenvalue of the empirical second moment matrix will be at least $1/2$.

