# OpenReview forum: "Contrastive estimation reveals topic posterior information to linear models"
_ICLR.cc/2021/Conference — Reject_

### Official Review · AnonReviewer4 · 2020-10-27
**Interesting analysis, but presentation could be improved**

**Rating:** 5
**Confidence:** 4

**Review:**

Summary:
Contrastive learning is applied in a semi-supervised setting with few training examples to provide features for a linear classifier. A theoretical analysis is provided to show that contrastive embeddings allow to predict using a linear transformation. The model allows to recover the topic structure of a corpus that is generated from a generative topic model.

Evaluation:

Overall, the paper provides a thorough theoretical analysis to prove that the approach allows to recover topic posterior information. However, I am not convinced of the impact of this analysis, and the experiments are not entirely convincing. Especially the chosen LDA baseline is weak and some parameter settings are not discussed. If the corpus is generated from an LDA generative model, LDA should be able to recover the topic information. A stronger Gibbs sampling variant would be more suitable in my opinion whereas other baselines like the BOW are not very informative.

To sum up, the objective of the paper is not entirely clear. While the theoretical analysis is interesting, I believe the presentation, clarity and writing of the paper should be improved. I suggest to provide a stronger motivation for the theoretical analysis with an outlook for future work where this analysis could be useful.

Strong points:
- Theoretical analysis
- Detailed proofs

Weak points:
- The analysis rests on the assumption that the corpus is generated from a certain generative process. This is a strong assumption. In topic modeling, this assumption is usually justified in that the resulting topics turn out to be interpretable. However, the approach in this paper does not recover any topics specifically. It is applied for classification where generative approaches are often not the preferred choice. So it seems to me like the advantage of the generative model, the interpretability, is stripped away, and the model is repurposed for classification.
- The experimental baselines are weak. Word2vec is not developed for the purpose of document classification, online VB does not have classification performance as a strong point and bare BOW is surely not suitable as a realistic baseline.
- Hard to follow at times, structure could be improved



Detailed comments:
- Eq. on p. 2: l is not explained
- Setup: A k-dimensional vector lies in the (k-1)-dimensional simplex, not the k-dimensional simplex
- The notation is slightly confusing. I would suggest to use the commonly used $\theta$ instead of $w$ for the topic distribution
- "By Bayes’ theorem we have that g^{\star}:= \exp(f^{\star}) satisfies the following:" Please give some more hints here about what you mean. Write that the Bayesian theorem is applied in the last equality and that the label y=0 makes the document parts independent. Also, why is y ommited in the last expression?
- independent conditioned -> independently conditioned
- all of these methods all of these methods -> all of these methods
- I find it confusing that the parameter $\alpha$ is divided by $K$ the number of topics. In FIgure 1 left, if this is the Dirichlet parameter $\alpha$ then this means you vary it between 1/20 and 10/20 or is this $\alpha/K$
- Experiment with LDA: you write that you use online VB, but you do not report the parameters such as batch size. Online VB is extremely sensitive to the batch size.
- If you compare classification performance I would suggest to rather compare to the Gibbs sampling version of LDA which usually will give better performance with enough samples. Scalability should not be an issue with the size of data sets that are used here.
- The two variants Direct NCE and Landmark NCE should be more explicitly introduced. At the moment, the Direct NCE introduction is hidden and hard to find. In the beginning you say your approach is to use landmarks and the Direct variant is rather suddenly introduced in a later part without discussing the differences and implications in much detail.

Update:

The authors have not fixed some of the errors I pointed out in my review. For example, they still refer to the k-dimensional probability simplex and did not address some of the other corrections. Since it is otherwise a promising paper, I suggest to carefully revise and resubmit. I will not however recommend a paper with errors for acceptance.

---

### Official Review · AnonReviewer1 · 2020-10-28
**Interesting paper**

**Rating:** 6
**Confidence:** 2

**Review:**

Summary:
This paper tries to learn a document level representation from document level contrastive estimation. The training task is try to predict where two half of a document are from the same document. The author proved the contrastive estimation reveals topic posterior information given the topic modeling assumptions. And in experiments, linear models can get relatively good performance.

Reasons for score:

Overall, I think it's a good paper. I like the idea of  using topic models as a way to represent the document level information, but it is disappointed to see that the proposed method doesn't provide as good performance as the simply averaging word embeddings.

Strong points:
1) Detailed proof and detailed experiments, compared with many baseline models.
2) Using topic modeling as a tool to understand representation learning is interesting.

Weak points:
1) Author doesn't provide complexity analysis for inference. For a new document, the speed of inference seems depends on number of landmark documents, which could be slow.
2) I don't understand why it's limited to linear models for the classification task. It seems we can use other models. If contrastive estimation can reveal topic posterior information to linear models, it can also reveal the information to non-linear models.
3) In the experiments section, the paper proposes a Direct-NCE method that has best empirical performance , which can be confusing for readers.

---

### Official Review · AnonReviewer2 · 2020-10-28
**Well written theoretical topic modeling paper**

**Rating:** 7
**Confidence:** 4

**Review:**

This submission considers contrastive learning approach to representation learning under topic modeling assumptions. It proves that the proposed procedure can recover a representation of documents that reveals their underlying topic posterior information in case of linear models. It is experimentally demonstrated that the proposed procedure performs well in a document classification task with very few training examples in a semi-supervised setting.

The idea behind the proposed procedure is to split randomly sampled documents into two parts and either keep them as is with the positive label or replace one of the parts with a randomly sampled document and assign the negative label. These data are then passed to the binary cross entropy loss which is optimised to find a transformation that assigns the probability of co-occurence of the two parts.

The main theoretical result comprises the population case, where infinite amount of data is sampled from the described generative model under an additional anchor word assumption, and proves that in such case the outcome of the proposed contrastive learning procedure is linearly related to all moments of the topic posterior up to a cerain degree. The finite sample case is futher analysed.

Synthetic experiments illustrate performance of the proposed procedure when the data is sampled from a topic model with varying degree of sparsity. It is observed, as expected, that the quality of topic recovery is better when the degree of sparsity is higher.

The algorithm is further compared on the AG news topic classification dataset with standard benchmarks and it is observed that the performance of the proposed procedure is good and overall comparable with word2vec, being slightly better in the lower sample size setting. The authors thereofre conclude that the proposed procedure can be interesting in semi-supervised learning applications with low amount of data.

Overall, the submission is a solide pice of work and well written. The results are theoretically sound and the observed improvement in the low sample size setting could be of interest in some challenging applications and might be worth further investigating. This line of research could be interesting to the theoretical topic modelling community.

typo p. 7: all of these methods all of these methods

---

> ### Author Response · Authors · 2020-11-16
> **Reply to AnonReviewer2**
>
> Thank you for your review.

---

### Official Review · AnonReviewer3 · 2020-11-01
**I am waiting for the author's response**

**Rating:** 6
**Confidence:** 3

**Review:**

Summary
This paper presents a new contrastive learning algorithm for document representation. The main idea is to generate pseudo labeled two texts, whether the texts are coming from the same document. To learn the discriminating function between the two texts, the learning algorithm minimizes the cross-entropy loss function between them. With the function, the authors suggest the embedding function for a document with selected landmark documents. The authors also show the learned function can be represented by combining the topic posterior distribution and topic likelihood distribution. Experiments show that the suggested learning algorithm can identify hidden topics from a synthetic dataset. And the authors also show the usefulness of the representation in semi-supervised learning by classification performance and visualization.

The main strength of this paper is suggesting a sound and straightforward learning algorithm to construct document representation. And the authors explain the relatedness of the representation function and topics in a document. I really like to read this part and I think researchers can extend their ideas from these insights.

What concerns me most is the lack of comparing existing document representations. The authors compare the suggested representation with models that do not explicitly make document representation (maybe except LDA). I think it is hard to say that this is a fair comparison. I agree that the authors do not need to show that the suggested one outperforms all other baselines. But there are several recent document representation algorithms (Logeswaran & Lee, 2018; Arora et al., 2019; Yurochkin et al., 2019). The authors explain why they do not show the results of them in the Related work section. But I think we can run the same experiments in Arora et al., 2019 with the suggested algorithm representation. I do not understand why their distributions are unclear for the experiment in this paper. Is it a semi-supervised regression not classification? I am interested in these questions, and please describe it in more detail.

Overall, I admit that the suggested algorithm is a sound method to construct the document representation. But I have some questions that I want to listen to the author’s responses.

Questions
- I think the landmark documents are important for constructing representation. What are the reasonable ways to choose landmark documents empirically? Does changing the size of the number of landmark documents affect the performance of downstream tasks?
- In semi-supervised learning experiments, Direct-NCE outperforms Landmark-NCE. Can we say that the landmarks are not useful for the downstream tasks?
- Can we recover topics that are word distributions from the representation?
- Is it a errata in Lemma 1? $$\psi(x)_k = P(x^{(1)} = x | w = e_k) / P(x^{(2)} = x') $$?

Reference
Yurochkin, Mikhail and Claici, Sebastian and Chien, Edward and Mirzazadeh, Farzaneh and Solomon, Justin M, Hierarchical Optimal Transport for Document Representation, NeurIPS 2019, http://papers.nips.cc/paper/8438-hierarchical-optimal-transport-for-document-representation

---

### Decision · Program_Chairs · 2021-01-07
**Final Decision**

**Decision:**

Reject

**Comment:**

This promising work proves that the proposed contrastive learning approach to representation learning can recover the underlying topic posterior information given standard topic modelling assumptions. The work provides detailed proof and detailed experiments. The analysis is interesting and yields interesting insights. However, the experimental results are somewhat weak by lacking comparison with more recent document representation work.

Pros:
- Good detailed proofs and experiments.
- Interesting idea of using topic modelling to understand representation learning.

Cons:
- The description of DirectNCE is somewhat hidden and could be better introduced in the paper.
- Experimental baselines are weak lacking a comparison to recent document representation work such as Arora et al. 2019.
- Stronger classification baselines could be incorporated.